# Pharmacist Knowledge and Perceptions of Homeopathy: A Survey of Recent Pharmacy Graduates in Practice

**DOI:** 10.3390/pharmacy10050130

**Published:** 2022-10-09

**Authors:** Jordin Millward, Kasidy McKay, John T. Holmes, Christopher T. Owens

**Affiliations:** 1Department of Pharmacy Practice, Kasiska Division of Health Sciences, Idaho State University, Pocatello, ID 83209, USA; 2Department of Family Medicine, Idaho State University, Pocatello, ID 83209, USA

**Keywords:** homeopathy, community pharmacy, pharmacy curriculum, patient counseling, complementary and alternative medicine, complementary and integrative health, surveys

## Abstract

Homeopathic products are available over the counter in many pharmacies in the United States and are popular among consumers, although there is no conclusive evidence of their therapeutic effects. Pharmacists are obligated to provide well-informed, evidence-based information on these products, but many graduates may not be receiving adequate training in this area. This report outlines the results of a survey assessing whether taking a focused elective course in complementary and integrative health (CIH) affects knowledge and perceptions regarding homeopathy. A 22-question survey was developed and distributed to graduates of Idaho State University College of Pharmacy. Responses on survey items were compared between those who had reported taking the CIH elective course and those who had not. Of the 475 pharmacists, 89 completed the survey (response rate of 18.7%). Pharmacists who had taken the CIH elective course reported being more comfortable answering patient questions (82% vs. 44%, *p* < 0.001), felt more able to make recommendations (75% vs. 36%, *p* < 0.001), and felt they could explain the proposed mechanism of action of homeopathic remedies to their patients (87% vs. 61%, *p* = 0.002). Those who took the elective course were also more likely to say that any benefits of homeopathy were due to the placebo effect (82% vs. 64%, *p* = 0.007). A significantly higher portion of respondents who had not taken the elective course indicated that they could benefit from further training on CIH topics when compared with those who had taken the elective course (85% vs. 51%, *p* = 0.02). There was no significant difference between groups with respect to their use of reliable resources (e.g., PubMed and Natural Medicines) vs. unreliable sources (other internet searches or personal anecdotes) when addressing CIH-related questions. These findings indicate that pharmacists with more focused training in CIH are more comfortable, confident, and knowledgeable when discussing homeopathy. Such education should be provided more broadly to students in colleges of pharmacy.

## 1. Introduction

Complementary and alternative medicine, now referred to as complementary and integrative health (CIH), is commonly used in the United States. Natural products, the most frequently utilized subset of CIH, were used by 17.7% of adults and 4.9% of children in 2012, according to a survey conducted by the National Center for Complementary and Integrative Health (NCCIH) [1]. Homeopathic products are one such natural remedy and were purchased by 6 million Americans and 200 million people around the world that same year [1]. The popularity of these products appears to have great staying power, despite limited clinical trial data supporting their efficacy and an implausible mechanism of action [2].

Homeopathy originated in Germany in the 1700s, purporting the ideas that “like cures like” and that the dilution of remedies increases their potency. When taken together, these ideas translate to the practice of consuming an extraordinarily dilute dose of a substance in an attempt to relieve the same symptoms that the substance would normally cause in higher doses. For example, a solution to treat colic in babies composed of belladonna is diluted in pure water by a factor of 10–12, resulting in one-part original substance per one trillion parts diluent. In many commonly used homeopathic remedies, it is likely that not one single molecule of active substance remains, and the end product is often indistinguishable from the diluent [3]. Common homeopathic brands such as Hyland’s and Boiron are available for sale over the counter in many retail pharmacies in the United States. Most of these remedies claim to treat self-limited conditions, such as cough, earache, or mild-to-moderate pain of various types, and are allowed to do so by current U.S. Food and Drug Administration policies [4].

Scientific evidence for the efficacy of homeopathic remedies is largely lacking. A 2017 meta-analysis of double-blinded, placebo-controlled, randomized trials of non-individualized treatment reported that 49 of the 75 eligible trials were considered to be at a “high risk of bias”, and 23 others were at “uncertain risk.” The remaining three trials considered to have reliable evidence were heterogeneous in the clinical conditions they targeted. Neither the pooled data from all extractable trials nor the data from the three reliable trials showed a significant benefit for any purported condition [5]. This reflects the evidence of earlier systematic reviews, which have consistently found that methodological flaws and the lack of independent replications of positive findings result in a body of low-quality, unreliable evidence that does not show homeopathic remedies to be clinically different from placebo [6].

Patients who choose to use homeopathic remedies do so largely on their own and without medical supervision. Pharmacists, especially those working in the community retail sector, are often the most accessible healthcare professionals to patients who are considering purchasing these products, and pharmacists in this setting are ethically obligated to provide patients with well-informed and impartial counseling and advice. However, a 2003 study found that out of 400 pharmacists surveyed, 81% felt they had inadequate skills and knowledge to counsel patients on herbal medicine, and 90% felt they needed more exposure to these kinds of therapies in their professional curricula [7].

Most schools and colleges of pharmacy in the U.S. provide some form of CIH education, and a 2017 survey found that most schools (58.5%) offer both elective and required course components related to CIH, while some offer only required content (33%) and others offer only elective courses (8.5%). Of the 117 required courses described, 17 focused exclusively on CIH topics. The most common method for teaching CIH is to incorporate CIH topics into required disease-state-based therapeutics courses [8]. However, many studies have indicated that CIH education may be ineffective or incomplete. A 2008 initiative by a college of pharmacy to incorporate CIH into regular academic modules resulted in only 30% of students feeling as though they had learned enough about CIH [9]. In general, pharmacists often report feeling that they lack resources regarding CIH, are unprepared to answer questions, and are unlikely to ask their patients about their use of CIH, and they lack education regarding herbal supplements as part of their professional curricula [10,11].

Like many other colleges and schools of pharmacy, Idaho State University College of Pharmacy currently has some CIH components integrated into the required curriculum. In addition, an elective course entitled Evidence-Based Complementary and Integrative Health has been offered for many years and is now open to all students in the university’s Kasiska Division of Health Sciences. Specific learning outcomes of the course include:−Appropriately evaluate and utilize the scientific literature to provide an evidence-based approach to patient care and pharmacy practice.−Provide effective counseling to patients and/or caregivers, including proper instructions for self-care and the safe and effective use of medications and devices.−Provide effective education to a variety of audiences, including patients, other health care professionals, and the lay public.

Graduates who have taken this elective should have a better understanding of the proposed mechanisms of a variety of CIH therapies, the data behind their safety and efficacy, and how to access the information they need to be able to make an appropriate clinical decision for a patient. The present survey had two aims: to examine whether pharmacists who took this elective course during their education at Idaho State University College of Pharmacy (1) considered themselves to be better prepared and more adequately trained to educate and counsel patients on homeopathic products, including making recommendations for their use based on available evidence, and (2) are in general more knowledgeable on the subject of homeopathy, including its purported mechanism of action and potential clinical applications.

## 2. Methods and Materials

A descriptive survey to analyze pharmacist knowledge and perceptions regarding homeopathy was designed and distributed to 475 alumni of Idaho State University who still had an active university email account. Approval was obtained by the Idaho State University Institutional Review Board. The study was created using Qualtrics^®^ Software (September, 2017; Provo, UT, USA) and emailed to College of Pharmacy alumni who had graduated within the last five years. Survey distribution was conducted between August 2017 and September 2017. Data were collected via Qualtrics over the course of the six weeks that the survey remained open. Email reminders were sent to respondents twice, once at 2 weeks and once at 4 weeks.

Survey items were developed after conducting a review of the literature. Five pharmacists not affiliated with the study pilot-tested each item for clarity. Items that caused confusion or were unclear in their intent were altered based on the feedback provided. The questionnaire consisted of four sections. The first section addressed the demographic data of respondents (age, sex, graduation year, practice state, and practice type). The second section focused on knowledge-based questions and included a knowledge item that asked respondents to select the correct definition of homeopathy from a multiple-choice list. The remaining questions in this section asked respondents whether or not they had taken the CIH elective while in school and instructed them to rate their own knowledge and comfort level discussing homeopathy with patients using a Likert scale. The third section addressed attitudes and perceptions related to homeopathic remedies and was composed of Likert-scale items that assessed perceived effectiveness, harmfulness, and value. These items were created using a previously validated CIH questionnaire as a guide [7]. The final section addressed pharmacists’ sources of CIH-related information and the perceived need for CIH education in pharmacy education.

Data from completed questionnaires were analyzed using SAS Studio University Addition statistical software version 3.71. The two groups that were examined and compared were those who had taken the CIH elective during their education at Idaho State University and those who had not. Chi-squared tests were used to analyze demographic and other categorical data, and ordinal logistic regression was used to analyze Likert-scale data. The results for knowledge- and perception-based questions are reported as numerical responses on a scale of 1–5, with 1 representing “strongly agree” and 5 representing “strongly disagree.” Likert-scale data were considered continuous for the analysis. Differences between the two groups were considered statistically significant if the calculated *p* value was less than 0.05.

## 3. Results

### 3.1. Baseline Characteristics

Of the 475 potential respondents, 89 completed the survey (18.7% response rate). Of these respondents, eight did not answer all the questions, and these surveys were not assessed as part of the final data analysis. This left the final total of respondents assessed at 81. About half (55%) of these respondents had taken the CIH elective offered at Idaho State University. Respondents’ mean age was 32.2 years (SD 6.6); 52% of respondents were male and 48% were female. Most respondents practiced in either a retail (32%) or hospital/inpatient (31%) setting.

Baseline characteristics were similar between respondents who had taken the CIH elective and those who had not. One notable exception is that those who had taken the elective were significantly more likely to be working in a retail setting (20 respondents, 44%; *p* = 0.0078), and those who had not taken the elective were more likely to be working in a hospital/inpatient setting (15 respondents, 42%; *p* = 0.06). Other analyzed demographic characteristics were not significantly different between the two groups (see Table 1).

### 3.2. Knowledge Base

In total, 75% of respondents were able to correctly identify the proposed mechanism of homeopathy from a five-item list. The difference was not statistically significant between those who took the elective and those who did not (82% vs. 66%; *p* = 0.167). Questions that dealt with perceived knowledge and comfort level did show significant differences between the two groups. Respondents who took the elective course were significantly more comfortable answering patient questions (*p* = < 0.001), felt more adequately trained to make recommendations (*p* < 0.001), and felt more able to explain the proposed mechanism of homeopathy to patients (*p* = 0.002).

### 3.3. Perceptions

Most perception-based questions did not differ significantly between respondents who took the CIH elective and those who did not. Regardless of whether they took the CIH elective course or not, the majority of respondents do not recommend homeopathic products to patients (75% of pharmacists who took the CIH course vs. 56% of pharmacists who did not, *p* = 0.287). Similarly, most respondents in both groups believe that the use of CIH therapies without scientific backing should be discouraged (60% of pharmacists who took the CIH course vs. 64% of those who did not, *p* = 0.89). Respondents who had taken the CIH elective during their education were significantly more likely to believe that most benefits seen from the use of homeopathy are due to the placebo effect, but the perception was high in both groups (82% vs. 64%, *p* = 0.007)

### 3.4. Education and Sources of Information

Regardless of whether or not respondents had taken the CIH elective, they agreed that CIH education should be required in a pharmacy student’s curriculum (82% of pharmacists who took the CHI elective course vs. 83% of those who did not, *p* = 0.382). Respondents were more likely to believe that they would benefit from further training in complementary and integrative medicine topics if they had not taken the elective course (85% of pharmacists who had not taken the CIH elective vs. only 51% of those who had, *p* = 0.022).

Respondents indicated they were most likely to obtain their information from the Natural Medicines Comprehensive Database (85.2%), followed by internet search (52%) and PubMed (37%). Some respondents also reported obtaining information about natural products from more subjective resources, including peers and colleagues (27%), and some reported reliance on a history of benefits/detriments to themselves or others (23%). Approximately 67% of respondents used at least one “unreliable” source of information that was not strictly evidence-based (e.g., internet search, peers and colleagues, and past history of benefits or other anecdotal reports). There was no significant difference in the use of “reliable” (e.g., PubMed and Natural Medicines) vs. “unreliable” sources of information between those who took the CIH elective and those who did not (*p* = 0.64). The main survey items and comparisons between respondent groups are shown in Table 2.

## 4. Discussion

Our study set out to determine whether taking an elective course in complementary and integrative health (CIH) significantly affected pharmacists’ knowledge and/or perceptions regarding homeopathy. It was theorized that any statistically significant differences in responses between those who took such a course and those who did not could provide insight into how to better structure education on this topic and may also indicate whether the current integration of CIH into the professional curriculum is adequately addressing these topics. The goal of core pharmacy curricula is to prepare graduates to be able to make informed recommendations and appropriately answer patient questions and concerns about prescription medications and over-the-counter products. Pharmacists should be able to make recommendations on safe and effective complementary therapies, and homeopathy, though controversial, is being recommended more and more to augment standard medical care in different countries and in a variety of fields [12,13].

Our sample consisted of mostly younger, recent graduates practicing in an inpatient or retail setting (large chain pharmacy), and most respondents practiced in Idaho. The survey reached only respondents with an active email address, so responses may have been biased toward more recent alumni who were more likely to be monitoring their university email address.

The findings indicate that pharmacists who took the CIH elective did indeed rate themselves more comfortable, confident, and capable of discussing homeopathic remedies with patients. These respondents were also much less likely to feel the need for further training in the subject of complementary and integrative health practices. Current education and training as part of the pharmacy curriculum appears to create graduates who are willing and able to discuss these common products and formulate informed, evidence-based recommendations regarding their use. The ability of respondents to identify the correct purported mechanism of homeopathy was not affected by taking the CIH elective, although this may have to do with the relative ease of selecting the correct definition from a multiple-choice question.

Pharmacists who graduated from Idaho State University are generally knowledgeable of and have an unfavorable opinion regarding homeopathy. Most do not recommend homeopathic products. General attitudes towards homeopathy were less affected by additional coursework beyond the normal required curriculum on the subject than perceived knowledge: e.g., respondents were just as likely to agree that “homeopathy is a threat to public health” whether they took the CIH elective course or not. Notably, the only perception-based outcome that was significantly affected by additional CIH education was “most benefits seen from the use of homeopathy are due to the placebo effect.” Those who took the elective course were more likely to agree with this statement. This may indicate that direct education regarding homeopathy is more likely to lead to a rejection of the proposed mechanism based on available evidence, as was provided by that specific course and not covered in the required curriculum. Most graduates believe that CIH should be a mandatory part of a pharmacy student’s curriculum, and this outcome was not affected by whether or not the respondent took the elective.

Most respondents used reliable sources of information to research complementary and alternative therapies, with the Natural Medicines Comprehensive Database and PubMed being chief among them, though more than half of respondents indicated they used simple internet searches to gather information, which may result in biased information and even misinformation, depending on the source utilized. Nearly a quarter of respondents used information from peers and colleagues, and a similar number used personal experience/anecdotal evidence to inform their recommendations. Taking the elective did not appear to make graduates less likely to use “unreliable” sources of information, such as internet searches, information from peers and colleagues, and the history of benefits to themselves or others. This may indicate a need for a greater focus on utilizing appropriate information resources during pharmacy education.

Our study has several limitations. While the response rate for the survey was in the expected range, the overall sample size was small, which limits the statistical power. Respondents were mostly recent graduates and may not be representative of all pharmacists who graduated from Idaho State University. The relatively recent average year of graduation (2015) also makes it difficult to assess whether time weakens the benefit of taking the CIH elective while in pharmacy school. There also may be some selection bias: respondents who feel more strongly about homeopathy or who took the elective may be more likely to take the survey. Finally, while baseline characteristics were mostly similar, there was a statistically significant difference in the practice site between those who took the elective and those who did not, with a significantly higher number reporting being in retail practice. This could skew results, as pharmacists who work in a retail setting are probably more likely to come into contact with homeopathy and to interact with patients asking questions or requesting recommendations about these therapies.

## 5. Conclusions

This research indicates that standard pharmacy curricula are likely adequate for addressing most issues dealing with complementary and integrative health, although taking an elective course focused on these types of therapies does appear to improve a pharmacist’s ability to communicate with patients more accurately and specifically regarding some aspects of homeopathic remedies in particular. This may indicate that education regarding the specifics of homeopathy is not adequately addressed in most core pharmacy curricula. Given the prevalence and popularity of homeopathy and its dubious mechanism of purported action, it behooves colleges and schools of pharmacy to critically evaluate how this type of content is covered in their curricula in order to ensure that their graduates are competent and prepared to address patients’ questions and make evidence-based recommendations related to this important group of over-the-counter products.

## Figures and Tables

**Table 1 pharmacy-10-00130-t001:** Demographics of Survey Respondents.

	Total (*n* = 81)	Took CIH Elective (*n* = 45)	Did Not Take CIH Elective (*n* = 36)	*p* Value
Age	Mean = 32	Mean = 32.5 (range 25–62)	Mean = 31.9 (range 23–42)	*p* = 0.80
Gender	52% male 48% female	44% male 56% female	57% male 43% female	*p* = 0.49
Practice site		20 (44%) retail (large chain)	6 (17%) retail	*p* = 0.008
		10 (22%) inpatient	15 (42%) inpatient	*p* = 0.06
		5 (17%) ambulatory care	6 (11%) ambulatory care	
		6 (13%) independent (small community)	5 (14%) independent	
		4 (9%) other	4 (11%) other	
Current Location		23 (51%) still in Idaho	18 (50%) still in Idaho	*p* = 0.92

**Table 2 pharmacy-10-00130-t002:** Survey Items.

Survey Item	Took Elective (*n* = 45)			Did NOT Take Elective (*n* = 36)			*p* Value
	Agree	Disagree	Neither	Agree	Disagree	Neither	
“I can explain the proposed mechanism of action of homeopathy to a patient.”	39 (87%)	4 (9%)	2 (4%)	22 (61%)	10 (28%)	4 (11%)	0.002
“I feel adequately trained to make recommendations for or against homeopathy in my daily practice.”	34 (75%)	4 (9%)	7 (16%)	13 (36%)	18 (50%)	5 (14%)	<0.001
“I am comfortable answering patient questions about homeopathy.”	37 (82%)	4 (9%)	4 (9%)	16 (44%)	15 (42%)	5 (14%)	<0.001
“I recommend homeopathic products to patients.”	5 (11%)	34 (75%)	6 (14%)	7 (19%)	20 (56%)	9 (25%)	0.287
“Most benefits seen from the use of homeopathy are due to the placebo effect.”	37 (82%)	1 (2%)	7 (16%)	23 (64%)	3 (8%)	10 (28%)	0.007
“There is therapeutic value in the use of homeopathy for some patients.”	17 (38%)	22 (49%)	6 (13%)	9 (25%)	15 (42%)	12 (33%)	0.593
“The use of complementary and alternative therapies without scientific backing should be discouraged.”	27 (60%)	8 (18%)	10 (22%)	23 (64%)	3 (8%)	10 (28%)	0.89
“Conventional medicine could benefit from the inclusion of homeopathic remedies as a part of standard practice.”	8 (18%)	27 (60%)	10 (22%)	8 (22%)	19 (53%)	9 (25%)	0.251
“Homeopathy is a threat to public health.”	14 (31%)	20 (44%)	11 (25%)	10 (28%)	11 (30%)	15 (42%)	0.89
“Complementary or alternative therapy education should be required in a pharmacy student’s curriculum.”	37 (82%)	3 (7%)	5 (11%)	30 (83%)	4 (11%)	2 (6%)	0.382
“I would benefit today from further training on complementary and alternative forms of medicine, including homeopathy.”	23 (51%)	6 (13%)	16 (36%)	27 (85%)	3 (8%)	6 (17%)	0.022

## Data Availability

The datasets generated during and/or analyzed during this study are not publicly available due to the use of anonymous survey data; however, datasets are available from the corresponding author on reasonable request.

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
