# Peer review of "Pharmacist Knowledge and Perceptions of Homeopathy: A Survey of Recent Pharmacy Graduates in Practice"

_pharmacy, 2022, doi:10.3390/pharmacy10050130_

Round 1
Reviewer 1 Report
It is a well-written paper of interest to this working with pharmacy education. The results are clearly presented although perhaps not surprising since one would expect that taking a specific course would (hopefully) benefit the graduates in their working life. Here are som comments.
The number stated in the introduction regarding the use of natural products is from 2012. Can more recent data be presented? Are the numbers increasing or decreasing over time?
Line 104: ”have a better understanding” - please rephrase since it is not shown that the have a better understanding. They should have a better understanding based on the course content but you don’t know that.
Line 107: ”The present review” - please rephrase, the article is not a review
Line 142: please clarify what type of logistic regression that was used. Was a multiple regression analysis considered?
In the abstract is says that ”Of the 427 pharmacists, 81 completed the survey (response rate of 17%)” but in the results (line 151) it says ”Of the 475 potential respondents, 89 completed the survey (18.7% response rate)”. Please provide the correct informationen in both places.
Line 186 ”seem”, should be replaced with ”seen”
References should be added in the discussion to put the results in relation to other studies.
Author Response
It is a well-written paper of interest to this working with pharmacy education. The results are clearly presented although perhaps not surprising since one would expect that taking a specific course would (hopefully) benefit the graduates in their working life. Here are som comments.
The number stated in the introduction regarding the use of natural products is from 2012. Can more recent data be presented? Are the numbers increasing or decreasing over time?
We utilized the most recent National Health Interview Survey data reported on the U.S. National Center for Complementary and Integrative Health (NCCIH) website. This was conducted in 2012. There was an update in 2017, but it only included use of meditation, yoga, and chiropractic and not natural products or supplements. The survey is due to be updated by NCCIH, and is likely coming soon, but these are to our knowledge the most reliable and recent national data available on the topic.
Line 104: ”have a better understanding” - please rephrase since it is not shown that the have a better understanding. They should have a better understanding based on the course content but you don’t know that.
Yes, and we do say "should have a better understanding" for the reasons you indicated.
Line 107: ”The present review” - please rephrase, the article is not a review
This has been rephrased to "the present survey"
Line 142: please clarify what type of logistic regression that was used. Was a multiple regression analysis considered?
We used ordinal logistic regression to analyze the dependent variable (level of agreement 1-5 to the various items) related to the main independent variable of interest (took the CIH elective course or not). We decided to treat the ordinal Likert scale data as continuous for the purpose of the analysis as indicated in the methods section. Multiple regression analysis could have been used to take into account other potential independent variables, but since our main interest was in the difference between respondents who took the elective course and those who didn't, we confined our analysis to ordinal linear regression.
In the abstract is says that ”Of the 427 pharmacists, 81 completed the survey (response rate of 17%)” but in the results (line 151) it says ”Of the 475 potential respondents, 89 completed the survey (18.7% response rate)”. Please provide the correct informationen in both places.
Thank you. That was an oversight on our part. The abstract and results section now agree and cite the raw number of 18.7% response --but with further detail provided in the results section.
Line 186 ”seem”, should be replaced with ”seen”
Done. Thank you.
References should be added in the discussion to put the results in relation to other studies.
This was a great observation and we have now included two additional references (#12 and #13 that cite the use of homeopathy in France and India). These include attitudes of oncologists, general practitioners, and public health professionals toward the use of homeopathy and these, we think, help to place the results of our findings with pharmacists into a greater context globally and with other healthcare professionals. Thank you.
Reviewer 2 Report
REPORT:
Pharmacist Knowledge and Perceptions of Homeopathy: A Survey of Recent Pharmacy Graduates in Practice.
BRIEF SUMMARY:
The aim: to examine whether pharmacists who took the elective course (Evidence-Based Complementary and Integrative Health) during their education at Idaho State university College of Pharmacy considered themselves to be better prepared and more adequately trained to educate and counsel patients on Homeopathic products, make recommendation based on available evidence, and if they are in general more knowledgeable on the subject of homeopathy.
The objective: a comparison of pharmacists who had taken the CHI elective with those who had not, for knowledge, attitudes/perceptions, sources of information, and perceived need for CIH education.
“This is an important field of study given the need for pharmacists to provide well-informed, evidence-based information to consumers, about the use of homeopathic remedies.”
GENERAL COMMENTS:
· Overall, well written with a style that is easy to read and understand
· Typographical errors, lines 186 and 192
SPECIFIC COMMENTS:
Aims: Lines 107-112:
As there appears to be more than one aim, itemise them into individual dot points
Study population: Lines 151-156:
Given – total respondents assessed = 81 (line154). Given – 52% male (line156). Demographic data is given for these male respondents; what about the other 48% of respondents? It would appear that the results presented are for the males only. There is a mention of a difference between groups in the Abstract (line 29) but not sure which groups? If the other 48% are identified, it could enable a comparison of findings.
Is this study only about male pharmacists?
Author Response
GENERAL COMMENTS:
- Overall, well written with a style that is easy to read and understand
- Typographical errors, lines 186 and 192
Thank you. The typos have been addressed.
SPECIFIC COMMENTS:
Aims: Lines 107-112:
As there appears to be more than one aim, itemise them into individual dot points
Thank you. We have added additional information and better delineated our survey aims as suggested. There are two specific aims that have been more clearly spelled out in lines 109-115.
Study population: Lines 151-156:
Given – total respondents assessed = 81 (line154). Given – 52% male (line156). Demographic data is given for these male respondents; what about the other 48% of respondents? It would appear that the results presented are for the males only. There is a mention of a difference between groups in the Abstract (line 29) but not sure which groups? If the other 48% are identified, it could enable a comparison of findings.
Is this study only about male pharmacists?
No, this survey when to a sample of recent pharmacist graduates, both male and female. We were not as clear as we should have been in the text and in the table in describing these demographic characteristics. We have made clarifications to indicate that 52% of our respondents were male an 48% were female. Our comparison groups were between those who took the elective course and those who did not (had the standard curriculum only). We did not draw comparisons between male and female respondents and did not intend to analyze differences in response based on gender, only on whether or not they had taken the elective course.
Thank you.
Reviewer 3 Report
The reviewed article concerns the knowledge and beliefs of graduates of pharmaceutical studies in the field of homeopathy. It is an interesting paper in the area that still raises doubts and controversies, despite many years of presence on the pharmaceutical market. In addition, previous studies concerning pharmacists showed their inadequate skills and knowledge on natural therapies, including homeopathy.
The main problem the Authors wanted to solve is whether the graduates of pharmaceutical studies are able to inform patients about homeopathic medicines, in accordance with the principles of evidence-based medicine and their knowledge gained during their studies, and then supplemented from reliable scientific sources. The statistical survey was based on a comparison between two groups of graduates, where the first group completed the compulsory CIH course and the second group completed the optional course.
Overall, the present research shows that standard pharmacy programs are likely to be suitable for addressing most CIH problems, although the optional (elective) courses appear to improve pharmacists' ability to communicate with patients in the area of CIH, including homeopathy.
For me, it is important to ask about the specific learning outcomes for the compulsory course. If they are very limited compared to the electives courses, the results of the study seem, at least in part, predictable. For me, a more interesting observation is a secondary conclusion that pharmacists typically use "unrealiable" sources of information, such as internet searches, information from colleagues and colleagues, and a benefit history for themselves or others."
In summary, some of the observations are interesting and can serve as inspiration to improve curricula in pharmacy. My main doubts concern the scientific aspects of the described research and its relevance to the scientific community.
Author Response
The reviewed article concerns the knowledge and beliefs of graduates of pharmaceutical studies in the field of homeopathy. It is an interesting paper in the area that still raises doubts and controversies, despite many years of presence on the pharmaceutical market. In addition, previous studies concerning pharmacists showed their inadequate skills and knowledge on natural therapies, including homeopathy.
Thank you. We agree that this is an interesting topic that has not been explored specifically using the survey format of recent pharmacy graduates we have employed.
The main problem the Authors wanted to solve is whether the graduates of pharmaceutical studies are able to inform patients about homeopathic medicines, in accordance with the principles of evidence-based medicine and their knowledge gained during their studies, and then supplemented from reliable scientific sources. The statistical survey was based on a comparison between two groups of graduates, where the first group completed the compulsory CIH course and the second group completed the optional course.
Overall, the present research shows that standard pharmacy programs are likely to be suitable for addressing most CIH problems, although the optional (elective) courses appear to improve pharmacists' ability to communicate with patients in the area of CIH, including homeopathy.
For me, it is important to ask about the specific learning outcomes for the compulsory course. If they are very limited compared to the electives courses, the results of the study seem, at least in part, predictable. For me, a more interesting observation is a secondary conclusion that pharmacists typically use "unrealiable" sources of information, such as internet searches, information from colleagues and colleagues, and a benefit history for themselves or others."
In summary, some of the observations are interesting and can serve as inspiration to improve curricula in pharmacy. My main doubts concern the scientific aspects of the described research and its relevance to the scientific community.
Thank you for your feedback and comments. We agree that there are some interesting observations here in our brief report that may be of interest to the educational community and your point is exactly right -- we want these findings to be a starting point for consideration of curricular updates. We felt the survey methodology was appropriate for the questions we were exploring and it is our hope that additional time will be spent in professional curricula to reinforce principles of evidence-based medicine, especially as it relates to complementary and integrative therapies such as homeopathy, where validated information is sometimes lacking.
Round 2
Reviewer 2 Report
Comments have been very well addressed, thank you.
Author Response
Thank you.